

# Derivation of the mean annual water-energy balance equation based on an Ohms-type approach

Xu Shan[1], Xingdong Li[1], Hanbo Yang[1]

[1]State Key Laboratory of Hydro-Science and Engineering, Department of Hydraulic Engineering, Tsinghua University,
Beijing 100084, China

*Correspondence to*: Hanbo Yang (yanghanbo@tsinghua.edu.cn)

**Abstract.** The Budyko hypothesis has been widely used to describe precipitation partitioning at the catchment scale. Many empirical and analytical formulas have been proposed to describe the Budyko hypothesis. Based on dimensional analysis and mathematic reasoning, previous studies have given an analytical derivation, i.e., the Mezentsev-Choudhury-Yang (MCY)

equation. However, few hydrological processes are involved in the derivation. Note that similar to electrical circuits and atmospheric motions, this study tried to give a new derivation of the Budyko hypothesis based on an analogy of the Ohms-type approach and the homogeneity assumption. The derived equation has the same form as the MCY equation but has a more physical explanation than the mathematic reasoning proposed in previous studies. In addition, under conditions without the homogeneity constraint, a more general expression is $E = \frac{P(b+kE_0)}{[P^n+(b+kE_0)^n]^{1/n}}$, where $E$, $E_0$ and $P$ are evaporation, potential

evaporation and precipitation, respectively, and $n$, $k$ and $b$ are constants.

## 1 Introduction

The mean annual water-energy balance equation describes the long-term relationship of actual evaporation ($E$) with precipitation ($P$) and potential evaporation ($E_0$) at the catchment scale. This equation is widely used in ecological, climatological, and socioeconomic applications (Greve et al., 2015). Additionally, this equation has been proved to be a

powerful tool to assess changes in catchment water balance as a function of climate change [Roderick and Farquhar, 2011; Yang and Yang, 2011; Renner et al., 2012; van der Velde et al.,2013; Greve et al., 2015).

Many attempts were made to formulate the mean annual water-energy balance according to observations from different catchments (Schreiber, 1904; Ol'dekop, 1911; Budyko, 1958; Pike, 1964). Based on previous studies, Budyko (1974) proposed a hypothesis on the mean annual water-energy balance, i.e., the Budyko hypothesis, which was expressed

mathematically as follows:

$$E = f(E_0, P),\qquad(1)$$

with the boundary conditions:

$$E \to P \text{ as } E_0 \to \infty$$





$E \to E_0$ as $P \to \infty$, (2)

which are commonly referred to as "dry condition" and "wet condition". Initially, the function was suggested without any parameters, indicating no capacity to control the impact of different catchment characteristics on the water-energy balance. Later, considering the effects of landscape characteristics, an adjustable parameter was introduced to describe the impacts of catchment characteristics on the water-energy balance (Choudhury, 1999; Zhang et al., 2001).

In addition, many studies have attempted to achieve an analytical equation based on mathematical reasoning. First, Bagrov (1953) introduced a derivative of the mean annual water-energy balance, $dE/dP=1-(E/E_0)^n$, and Mezentsev (1955) assumed $m=(n+1)/n$, giving a modification of $dE/dP=[1-(E/E_0)^n]^m$ and obtaining an integration of

$$E = PE_0/(P^n + E_0^n)^{1/n} .$$ (3)

However, the meaning of $m=(n+1)/n$ was not given by Mezentsev (1955). Then, Fu (1981) assumed that the derivative of $E$ with respect to $P$ (or $E_0$) could be expressed as a function of the variables $E_0 - E$ and $P$ (or $P - E$ and $E_0$), i.e.,

$$\frac{\partial E}{\partial P} = f(E_0 - E, P) ,$$ (4)

$$\frac{\partial E}{\partial E_0} = f(P - E, E_0).$$ (5)

Furthermore, he derived one analytical solution by dimensional analysis and mathematical reasoning (Fu, 1981; Zhang et al., 2004) as follows:

$$\frac{E}{P} = 1 + \frac{E_0}{P} - [(1 + (\frac{E_0}{P})^w)]^{1/w},$$ (6)

Yang et al. (2008) suggested a more general assumption that $E$ can be described as an implicit function of $P$, $E_0$ and $E$, i.e., $E=E(P, E_0, E)$ (equation (5) in Yang et al., 2008), together with the boundary conditions, namely, a 0-order boundary condition similar to equation (2) and a 1-order boundary condition as follows:

$$\begin{cases} \dfrac{\partial E}{\partial P} = 0, & \text{at } P/E_0 \to \infty, \text{ or } E = E_0 \\[2mm] \dfrac{\partial E}{\partial E_0} = 0, & \text{at } E_0/P \to \infty, \text{ or } E = P \\[2mm] \dfrac{\partial E}{\partial P} = 1, & \text{at } P \to 0, E_0 \neq 0 \\[2mm] \dfrac{\partial E}{\partial E_0} = 1, & \text{at } E_0 \to 0, P \neq 0 \end{cases}$$ (7)

Furthermore, Yang et al. (2008) analytically derived a solution for the Budyko hypothesis that was similar to the formula derived by Mezentsev (1955) and suggested by Choudhury (1999) (equation (3)) and was therefore called the Mezentsev-Choudhury-Yang (MCY) equation. Recently, Zhou et al. (2015) gave a general derivation of all kinds of Budyko functions by introducing a generator function:



$$g(\varphi) = \frac{\frac{\partial E}{\partial P}}{\frac{\partial E}{\partial E_0}} \frac{P}{E_0} = \frac{F(\varphi) - \varphi F\prime(\varphi)}{\varphi F\prime(\varphi)}, \tag{8}$$

where $\varphi = E_0/P$ and $F(\varphi) = E/P$. Then, they obtained the MCY equation by choosing $g(\varphi) = \varphi^n$ and solving equation (8). Notably, the differential equations proposed in those studies are not very rigorous and do not reflect sufficient hydrological understanding.

In physics, the hydrological cycle shapes energy balances and interacts strongly with atmospheric motion and transport (Kleidon et al, 2013). Fluxes displayed in the hydrological cycle, such as evaporation and precipitation, could be described by thermodynamics. Accordingly, thermodynamic principles, such as the principle of maximum entropy production (MEP) (McDonnell et al., 2007; Kleidon and Schymanski, 2008; Kleidon, 2009, 2010 a,b; Zehe and Sivapalan, 2009; Schaefli et al., 2011) and Carrot Limit (Kleidon et al, 2013), are widely used to understand the hydrological cycle. Kleidon and Schymanski

(2008) reviewed the hydrological applications of MEP and proposed the expressions for entropy production. Wang et al. (2015) introduced their expressions to study catchment water balance and developed a two-parameter equation approaching the Budyko hypothesis, as follows:

$$\frac{E}{P} = \frac{1 + \varphi\varepsilon - \varepsilon + \varphi\frac{E_0}{P}\sqrt{\left(1 + \varphi\varepsilon - \varepsilon + \varphi\frac{E_0}{P}\right)^2 - 4\varphi\varepsilon(1 + \varphi - \varepsilon)\frac{E_0}{P}}}{2\varepsilon(1 + \varphi - \varepsilon)}, \tag{9}$$

where $\varepsilon$ represents the initial evaporation ratio and $\varphi$ represents the ratio of the continuing evaporation conductance to the

runoff conductance. Zhao et al. (2016) further derived a general catchment water balance expression unifying catchment water balance equations at different time scales. However, Westhoff et al. (2016) pointed out that, the results of Wang et al. (2015) had some contradictions with Westhoff and Zehe (2013).

Thus, in this paper, focusing on the subsequent transportation processes of the precipitated water over a certain catchment; we define a catchment network and assume that fluxes (including vapor transportation and phase transition) can be estimated

according to an Ohms-type approach. Furthermore, we propose a model for water vapor transportation in the catchment network and derive the mean annual water-energy balance equation. Section 2 gives the basic assumptions and a conceptual framework, Section 3 gives the main derivation, and the discussion and conclusions are given in Section 4 and Section 5, respectively.

## 2 Ohms-type approach

In a catchment, there are two kinds of water phase transition, namely evaporation, condensation of the water vapor to precipitation. Water vapor enters a certain catchment through atmospheric motion, and then condenses as the precipitation. Part of the liquid water would evaporate as evaporation, the other part of the liquid water will confluence as runoff. Subsequently, water vapor from evaporation can be precipitated in the same catchment or transported to other catchments





due to atmospheric motion. We assume that the water phase transition and transportation can be approached using an Ohms-type law.

## 2.1 Definitions and assumptions

First, we focus on the phase transition and transportation of water and propose a catchment network(Figure 1). As shown

in Figure 1, Catchment $A_1$ is a chosen catchment for water balance analysis. $P_1$ is the atmospheric water vapor that forms precipitation over Catchment $A_1$. $E_1$ is the evaporation of Catchment $A_1$ that then precipitates on Catchment $A_1$ and other catchments, which are denoted by Catchment $A_{2,j}$ ($j$=1, 2, 3,…). The precipitation originating from $E_1$ and falling on Catchment $A_{2,j}$ is denoted by $P_{2,j}$. The sum of $P_{2,j}$ ($j$=1, 2, 3,…) is denoted by $P_2$. Notably, $P_{2,j}$ is just the part of the precipitation falling on Catchment $A_{2,j}$. Next, $P_{2,j}$ ($j$=1, 2, 3,…) partitions into two parts, namely, evaporation $E_2$ and runoff

$R_2$. Similarly, $E_2$ is all the evaporation originating from $P_{2,j}$ ($j$=1, 2, 3,…), and it precipitates on Catchment $A_{3,j}$ ($j$=1, 2, 3,…). Here, we track only the transformation and transportation of the vapor $P_1$, and these processes can be simplified as shown in Figure 2. Finally, $P_1$ is divided into runoff $R_i$ ($i$=1, 2, ..., $n$) and evaporation $E_n$. $E_1$ is part of $P_1$, i.e. $E_1 = k_1 P_1$, with $k_1 < 1$. Similarly, $E_2 = k_2 E_1 = k_1 k_2 P_1$, with $k_2 < 1$. Finally, $E_n = \prod_{i=1}^{n} k_i P_1$, with $k_i < 1$. Therefore, when $n \to \infty$, there is $E_n \to$ 0 and $P_1 = \sum_{i=1}^{n} R_i$. In other words, the initial water vapor of precipitation $P_1$ completely transforms into runoff after

numerous precipitation-evaporation-precipitation transformations.

The generalized flux is defined as the potential difference divided by the resistance and is a function of flux. That is, all the generalized fluxes here are driven by some kind of potential difference or potential gradient. In addition, some essential assumptions are given as:

**Assumption 1**: The mathematical form of the generalized flux is a positive single-value increasing function with respect to

the absolute amount of water flux within the water movement process during a certain period.

**Assumption 2**: The mathematical form of the generalized flux does not vary with different water movement processes within a catchment and between catchments.

**Assumption 3**: The potential of liquid water is assumed to be zero.

Hence, in a certain catchment, water vapor condenses to precipitation, and then, part of the precipitation evaporates, while

runoff is formed from the other part of the precipitation. Over a long duration and by ignoring the water storage change, the catchment water balance can be expressed as

$$E_1 = P_1 - R_1 , \tag{10}$$

where $E_1$ is the evaporation from Catchment $A_1$, $P_1$ is the water vapor which will form as precipitation over Catchment $A_1$, and $R_1$ is the runoff from Catchment $A_1$.

The resistance of the water vapor movement or transportation process $\eta$ can be expressed as

$$\eta = \frac{\Delta U}{f(x)} , \tag{11}$$





where $\Delta U$ represents the potential difference and $f(x)$ represents the generalized flux, which is defined as a function of flux (such as precipitation, evaporation and runoff, denoted by $x$) in the transportation and transformation processes. However, equation (11) is a trivial relationship similar to Ohms Law in electromagnetics (where $\Delta U$ represents voltage, $f(x)$ represents electrical current, and $\eta$ represents electrical resistance) and Darcy's Law in hydrodynamics (where $\Delta U$ represents

water head loss, $f(x)$ represents discharge, and $\eta$ represents a resistance parameter).

## 2.2 Physical reasoning

We focus on the precipitation partition over Catchment $A_1$. In Figure 3, Node B represents Catchment $A_1$, and Node A represents the atmosphere over Catchment $A_1$. Catchment $A_2$ represents a group of catchments where the water vapor from $E_1$

can precipitate. Similarly, Node D represents Catchment $A_2$, and Node C represents the atmosphere over Catchment $A_2$. $P_1$ (gaseous state) is the water vapor that precipitate on Catchment $A_1$. Over a long duration, the net water vapor flux transported from Node A to Node B equals $P_1-E_1$ ($R_1$), that from Node A to Node C equals $E_1$, and that from Node C to Node D equals $R_2$ (liquid state). Consequently, according to the definition, the generalized flux between Nodes $A$ and $B$ is $f(R_1)$, that between Nodes A and C is $f(E_1)$, and that between Nodes C and D is $f(R_2)$.

1) Net water vapor flux is transported into Node A via Path $P_1$ in the form of total precipitation.

2) Water exists in a gas state in Nodes A and C and a liquid state in Nodes B and D. Thus, Path A→B represents the phase transition of vapor in the process of condensation. Path A→C represents the vapor transportation driven by the potential difference $U_2 - U_1$.

3) The potential difference between B and D is zero since the potential of liquid water is zero. The potential difference

driving the phase transition of condensation is equal to the potential difference between the vapor and liquid water. The potential difference between A and D ($\Delta U_{AD}$) and equals that between A and B ($\Delta U_{AB}$), since the potentials of B and D are zero.

Two additional corollaries are as follows:

(a) **Corollary 1**: There are similar resistances during Path A→B and Path C→D since they are the chase transition from

vapor to liquid. Therefore, $\eta_1$ and $\eta_3$ have similar values when assuming the same temperature.

$$\eta_1 = \eta_3 , \tag{12}$$

(b) **Corollary 2**: There are sufficient occurrences of water transportation as $n \to \infty$, which lead to $\eta_{AB} = \eta_{CD}$. Note that $\eta_{AB} \neq \eta_1$. Here, $\eta_{AB}$ is the net resistance of all the possible roads between Node A and Node B, including Path A→B and Path A→C→D→B. Similarly, $\eta_{CD}$ is the resistance of all possible roads between Nodes C and D.

Thus, we have a general equation:

$$\eta_{AD} = \eta_{CD} + \eta_2 , \tag{13}$$





According to equation (11), the resistances can be estimated as $\eta_{AB} = \frac{\Delta U_{AB}}{f(P_1)}$ and $\eta_{AD} = \frac{\Delta U_{AD}}{f(E_1)}$. Consequently, the equation $\eta_{CD} = \eta_{AB}$ leads to

$$\eta_{CD} = \eta_{AB} = \frac{\Delta U_{AB}}{f(P_1)}, \tag{14}$$

Because $\Delta U_{AD} = \Delta U_{AB}$, we can obtain

$$\eta_{AD} = \frac{\Delta U_{AD}}{f(E_1)} = \frac{\Delta U_{AB}}{f(E_1)}, \tag{15}$$

According to the boundary condition, $E_1 \rightarrow E_0$ and $R_1 \rightarrow \infty$ when $P_1 \rightarrow \infty$. This condition indicates that much more water is draining via Path $R_1$ than evaporating via Path $E_1$, which means $\eta_1 \ll \eta_2$ and $\eta_3 \ll \eta_2$. In addition, the resistance of $\eta_{CD} < \eta_3$ since $\eta_{CD}$ is a result of the parallel of $\eta_3$ and the resistance of the remaining part. Thus, $\eta_2 + \eta_{CD} < \eta_2 + \eta_3$. The boundary condition $P_1 \rightarrow \infty$ yields that $\eta_{AD} = \eta_2 + \eta_{CD} \rightarrow \eta_2$, i.e.,

$$\eta_{AD} = \eta_2, \tag{16}$$

Substitution of equation (15) into equation (16) leads to

$$\eta_2 = \frac{\Delta U_{AB}}{f(E_1)} = \frac{\Delta U_{AB}}{f(E_0)}, \tag{17}$$

Substitution of equations (13), (14) and (17) into equation (13) leads to

$$\frac{1}{f(E)} = \frac{1}{f(E_0)} + \frac{1}{f(P)}, \tag{18}$$

**3 Derivation of the mean annual water-energy balance equation**

According to Hankey et al. (1971), the homogeneity of the Budyko function is expressed as:

$$E = E(P, E_0), \tag{19}$$

Garrison (2017) showed that equation (19) follows the Euler relation:

$$E = \frac{\partial E}{\partial P} P + \frac{\partial E}{\partial E_0} E_0, \tag{20}$$

The Euler relation indicates the homogeneity of the Budyko function. Combining the Euler relation with the result of the Ohms-type model (18), we shall be able to derive a Budyko function.

Note that if using $F(x) = \frac{1}{f(x)}$ to substitute $f(x)$ in (18), we can obtain:

$$F(E) = F(P) + F(E_0), \tag{21}$$

Taking partial derivatives of both sides with respect to $P$ and $E_0$ obtains the following:





$$\begin{cases} F'(E)\frac{\partial E}{\partial P} = F'(P) \\ F'(E)\frac{\partial E}{\partial E_0} = F'(E_0) \end{cases} \Rightarrow \begin{cases} \frac{\partial E}{\partial P} = \frac{F'(P)}{F'(E)} \\ \frac{\partial E}{\partial E_0} = \frac{F'(E_0)}{F'(E)}, \end{cases}$$

(22)

Substituting the two partial derivatives from equation (18) into equation (21), we obtain:

$$F'(E)E = F'(P)P + F'(E_0)E_0,$$

(23)

By comparing equations (21) and (23), a solution for $F(x)$ can be easily determined if $F'(x)x = mF(x)$:

$$F(x) = cx^m \Rightarrow f(x) = \frac{1}{c}x^{-m},$$

(24)

We assume that $f(x)$ must be a positive single-value increasing function; thus, $c > 0$ and $m < 0$.

To simplify the result:

$$Ff(x) = ax^n,$$

(25)

where $a, n > 0$.

Then, we can substitute equation (25) into equation (18) and obtain

$$\frac{1}{E^n} = \frac{1}{P^n} + \frac{1}{E_0^n},$$

(26)

Equation (26) can be transformed into $E = \frac{PE_0}{(P^n + E_0^n)^{1/n}}$, which has the same form as the MCY equation.

## 4 Discussions

### 4.1 Generalized flux

Flux is generally defined as the quantity that passes through the surface (Maxwell, 1873). There are several forms of flux, such as momentum flux ($N \cdot s \cdot m^{-2} \cdot s^{-1}$), heat flux ($J \cdot m^{-2} \cdot s^{-1}$), mass flux ($kg \cdot m^{-2} \cdot s^{-1}$), and electric flux ($N \cdot C^{-1}$). Flux can be estimated as the potential difference divided by resistance, and for example in Darcy's law, the water flux ($Q$) can be estimated as $Q = J/r$, where $J$ is the hydraulic slope and $r$ is the resistance. An alternate form of Darcy's law is $v = J/r'$, where $v$ is the velocity (or the flux density) and $r' = r/A$ (where $A$ represents sectional area). In this study, we defined the

generalized flux as a function of the flux, i.e., $f(x)$, where $x$ represents some form of flux. The generalized flux can be used to describe a more general relationship between fluxes and potential differences. For example, under turbulent conditions, $v^2 + bv = J/K_1$ (Forchheimer, 1901), i.e., the generalized flux $f(x)=x^2 + bx$. In other words, flux has a linear relationship with potential difference, while generalized flux can describe a nonlinear relationship between a given flux and potential difference. Equation (25) defines the generalized flux of water flux at the catchment scale, and parameter $n$ was reported

from 0.4 to 3.8 (with a mean of 1.3) for 210 catchments across China (Yang et al., 2014). This equation indicates a nonlinear relationship, except for $n=1$. In addition, the mean value of 1.3 is larger than 1, and the catchment water balance is



speculated to have some similarity with the behavior of groundwater flow. Remarkably, some catchments have an $n$ value of less than 1. Therefore, the mechanism behind the nonlinear relationship needs further study.

**4.2 Physical understanding of the Budyko hypothesis**

According to the Ohms-type approach, the partition of precipitation into evaporation and runoff is dependent on the two resistances $\eta_1$ and $\eta_2$. Resistance $\eta_1$ is related to the condensation processes of water vapor. Resistance $\eta_1$ can be estimated as $\eta_1 = \Delta U_{AB}/f(P-E)$. In this study, we assumed that the potential of liquid water is zero, so the potential of water vapor is $\lambda$ under the simplest condition ($n = 1$), $\eta_1 = \lambda/a$, whereas when $n$ does not equal 1, $\eta_1$ has a sophisticated form similar to Darcy's law under turbulent conditions. Additionally, resistance $\eta_2$ can be estimated as $\eta_2 = \Delta U_{AB}/f(E_0)$ according to equation (11). Remarkably, there is an implicit assumption that $f(x)$ is homogeneous in the horizontal and vertical directions. If $f(x)$ is not homogeneous, we denote $\varphi(E_0) = \Delta U_{AB}/\eta_2$, and we can speculate $\varphi(x) = b + kf(x)$ (where $b$ and $k$ are constants) since $\varphi(x)$ should have the same dimension as $f(x)$. Thus, $\frac{1}{E^n} = \frac{1}{P^n} + \frac{1}{(b+kE_0)^n}$, i.e.,

$$E = \frac{P(b+kE_0)}{[P^n+(b+kE_0)^n]^{1/n}},\qquad(27)$$

When $b = 0$, equation (27) can be simplified as $E = \frac{kPE_0}{[P^n+(kE_0)^n]^{1/n}}$, which is the same as that proposed by Zhou et al. (2015) (equation (21)).

This study proposed a catchment network in which the initial water vapor precipitated over Catchment $A_1$ can be completely transformed into runoff after infinite iterations of the precipitation-evaporation process. In the Ohms-type approach, as shown in Figure 3, we assumed that the resistances $\eta_1$ and $\eta_2$ have the same forms, which means that the generalized flux $f(x) = ax^n$ has the same values of $a$ and $n$ for Catchments $A_1$ and $A_2$. As is well known, $n$ represents the catchment characteristics (Yang et al., 2008). Therefore, Catchments $A_1$ and $A_2$ have similar characteristics under these conditions. The vapor that evaporated from Catchment $A_1$ possibly precipitated over the adjacent catchments, which leads to similar $n$ values. Under the condition that Catchments $A_1$ and $A_2$ have different characteristics, a large difference in $n$ occurs, which will leads to a more complicated form. Therefore, further study on the Ohms-type approach is still required.

**4.3 Derivation of the mean annual water-energy balance equation**

Equation (18) can be considered a new constraint on the water-energy balance. The form of the generalized flux determines the mathematical form of the mean annual water-energy balance equation. According to equation (22),

$$g(\varphi) = \frac{F'(P)P}{F'(E_0)E_0},\qquad(28)$$

where $g(\varphi)$ is the generator function in Zhou et al. (2015) and $\varphi = \frac{E}{P}$. Assuming that $F'(P)$ and $F'(E_0)$ have the same mathematical form, only the MCY equation satisfies the constraint of equation (18). Thus, we can draw a similar conclusion



to that by Zhou et al. (2015), i.e., the MCY function is the best function among the existing Budyko functions to improve our understanding of hydrological cycles.

This study derived the mean annual water-energy balance equation by using the Euler relationship to represent the homogeneity of the Budyko function. Additionally, the derivation can be drawn following Zhou et al. (2015), which

introduced homogeneity by using a generator function, and additional details are provided in the Appendix.

## 5 Conclusions

Previous studies have analytically derived the mean annual water-energy balance mainly by mathematical reasoning, such as Fu (1981), Yang et al. (2008), and Zhou et al. (2015). Remarkably, this study proposed a catchment network to describe water transportation and transformation and assumed water fluxes can be estimated by using an Ohms-type approach.

Furthermore, the MCY equation $E = \frac{PE_0}{(P^n + E_0^n)^{1/n}}$ can be derived to approach the Budyko function according to the homogeneity assumption expressed as the Euler relationship (Hankey et al., 1971) or a generator function (Zhou et al., 2015). In addition, without the homogeneity assumption, this study derived a general form $E = \frac{P(b+kE_0)}{[P^n + (b+kE_0)^n]^{1/n}}$, where $b$ and $k$ are constants.

## Acknowledgments

This research was partially supported by funding from the National Natural Science Foundation of China (Grant Nos. 51622903 and 41661144031), the National Program for Support of Top-notch Young Professionals, and the Program from the State Key Laboratory of Hydro-Science and Engineering of China (Grant No. 2017-KY-01).

## Appendix. Derivation of the MCY equation based on the generator function

Based on equation (18), the derivatives of $\frac{1}{f(E)}$ with respect to $P$ and $E_0$ can be expressed as follows:

$$\frac{\partial(\frac{1}{f(E)})}{\partial E_0} = -\frac{1}{f(E_0)^2}\frac{\partial f(E_0)}{\partial E_0},$$ (A1)

$$\frac{\partial(\frac{1}{f(E)})}{\partial P} = -\frac{1}{f(P)^2}\frac{\partial f(P)}{\partial E_0},$$ (A2)

since $\frac{\partial(\frac{1}{f(P)})}{\partial E_0} = \frac{\partial(\frac{1}{f(E_0)})}{\partial P} = 0$. In addition, the derivatives of $\frac{1}{f(E)}$ can be expressed as





$$\frac{\partial\left(\frac{1}{f(E)}\right)}{\partial E_0} = -\frac{1}{f(E)^2}\frac{\partial f(E)}{\partial E}\frac{\partial E}{\partial E_0}, \tag{A3}$$

$$\frac{\partial\left(\frac{1}{f(E)}\right)}{\partial P} = -\frac{1}{f(E)^2}\frac{\partial f(E)}{\partial E}\frac{\partial E}{\partial P}, \tag{A4}$$

According to equations (A1) and (A3) and equations (A2) and (A4), we can obtain

$$\frac{1}{f(E)^2}\frac{\partial f(E)}{\partial E}\frac{\partial E}{\partial E_0} = \frac{1}{f(E_0)^2}\frac{\partial f(E_0)}{\partial E_0}, \tag{A5}$$

$$\frac{1}{f(E)^2}\frac{\partial f(E)}{\partial E}\frac{\partial E}{\partial P} = \frac{1}{f(P)^2}\frac{\partial f(P)}{\partial E_0}, \tag{A6}$$

Dividing equation (A5) by equation (A6) obtains

$$\frac{\partial E/\partial P}{\partial E/\partial E_0} = \frac{\partial f(P)/\partial P}{\partial f(E_0)/\partial E_0}\cdot\frac{f(E_0)^2}{f(P)^2}, \tag{A7}$$

Based on the Budyko hypothesis, a general function $g(\emptyset)$ is given by Zhou et al. (2015) as

$$g(\emptyset) = \frac{\partial E/\partial P}{\partial E/\partial E_0}\cdot\frac{P}{E_0}, \tag{A8}$$

where $\emptyset = E_0/P$, and $0 < g(\emptyset) < +\infty$, $g'(\emptyset) > 0$. Therefore, based on equations (A7) and (A8), we can obtain another form of the general function given by

$$g(\emptyset) = \frac{\partial E/\partial P}{\partial E/\partial E_0}\cdot\frac{P}{E_0}, \tag{A9}$$

A general property of $g(\emptyset)$ was derived by Zhou et al. (2015) as follows since the constraints on evaporation due to available water and energy are symmetrical.

$$g(\emptyset)\cdot g\left(\frac{1}{\emptyset}\right) = 1, \tag{A10}$$

We assume $h(x)$ is an odd function, and $g(\emptyset)$ can be represented as

$$g(\emptyset) = e^{h(ln\emptyset)} \tag{A11}$$

and an easy form of $h(x)$ is $h(x) = kx, k > 0$. Thus, we can obtain one form of the general function:

$$g(\emptyset) = \emptyset^n \tag{A12}$$

where $n = b_j > 0$ $and$ $n \in R$. Therefore, we can obtain

$$\left(\frac{E_0}{P}\right)^{n+1} = \frac{\partial f(P)/\partial P}{\partial f(E_0)/\partial E_0}\cdot\frac{f(E_0)^2}{f(P)^2} \tag{A13}$$

and the case:





$$\frac{\partial f(P)}{\partial P} \cdot \frac{P^{n+1}}{f(P)^2} = \frac{\partial f(E_0)}{\partial E_0} \cdot \frac{E_0^{n+1}}{f(E_0)^2} \tag{A14}$$

As $f$ is only the function with respect to one variable, the left hand side of equation (A14) can be equal to a constant number as follows:

$$\frac{\partial f(P)}{\partial P} \cdot \frac{P^{n+1}}{f(P)^2} = const = a \tag{A15}$$

that is,

$$\frac{\partial f(x)}{\partial x} \cdot \frac{x^{n+1}}{f(x)^2} = const = a \tag{A16}$$

$$d\left(\frac{1}{f(x)}\right) = d\left(\frac{a}{nx^n}\right) \tag{A17}$$

$$f(x) = \frac{nx^n}{a} \tag{A18}$$

Based on equations (A18) and (A1), the MCY equation is represented as:

$$\frac{1}{E^n} = \frac{1}{P^n} + \frac{1}{E_0^n} \tag{A19}$$

or the form we are familiar with:

$$E = \frac{PE_0}{(P^n + E_0^n)^{1/n}} \tag{A20}$$

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



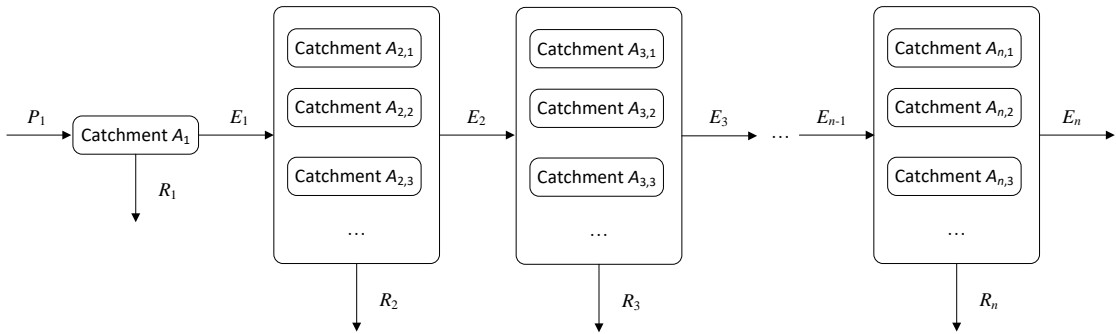

**Figure 1: A conceptual diagram for water vapor transportation and transformation processes within a catchment network.**

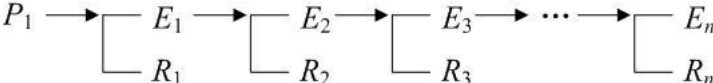

**Figure 2: Water vapour transformation within a catchment network.**

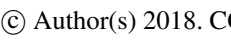



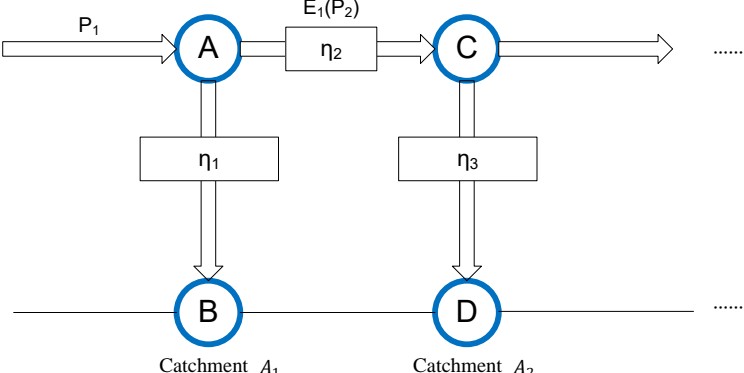

**Figure 3: One conceptual network model describing water transportation within a catchment and between catchments over a long duration. The arrows represent the path and direction of water movement. Note that Catchment $A_2$ represents a group of catchments in which the water vapor that evaporated from Catchment $A_1$ might precipitate.**

