# Peer review of "Derivation of the mean annual water-energy balance equation based on an Ohms-type approach"

_Hydrology and Earth System Sciences, 2018_

## Referee Comment (RC1) · Anonymous Referee #1 · 24 Dec 2018

The paper aims for a physically based derivation of the mean water and energy balance equation. The authors use a description of water vapor transfer between catchments and describe the fluxes in a flux gradient approach. Imposing the Budyko hypothesis then yields the well known Mezentsev-Choudhury-Yang equation. I think that a general derivation of the Budyko or the MCY equation is of high interest for hydrological research and thus of interest for HESS. However, one of aims of this paper is a rigorous derivation which reflects hydrological understanding. To be honest, I find it difficult to understand the reasoning which form the basis for the derivation.

What I do not understand is the framework of water vapor transfer between catchments (illustrated in the figures). Figure 1 and 2 show a moisture transfer from one catchment to the next (downwind?) where the input of the next catchment is set by the evaporation

of the first. This then leads to their statement that after n catchments there is no water left and E = 0. However, in reality there are also other sources of vapor which can contribute to precipitation in catchment 2 and these are being neglected. So this framework is not intuitive to me. The main derivation is illustrated with figure 3 where 4 different nodes are introduced. With corollary 2 it is stated that the resistance $n_{AB} \neq n_1$ arguing that there are other possible routes between the node Atmosphere A and the catchment node B, namely atmosphere A to atmosphere C to catchment D to catchment B. I did not understand this water transfer between the two catchments. I am also not sure if these assumptions and the ones in stated in section 2.1 are actually relevant for the derivation described in section 3. Therefore I recommend major revisions which should particularly improve the description to enable the reader to better understand how considering hydrological processes lead towards the MCY equation.

Further remarks:

abstract: "homogeneity assumption" should be described more specific to the paper

End of abstract, L15: There is no conclusion provided. Please explain what your results imply.

P5L2-5: it is unclear why this is mentioned here

P6L18: Garrison 2017 not in bibliography

P8L10: $\phi(E_0) =$ ; while $\phi = E/P$ on L27 ; unclear why the symbols is used for different meanings

P9L1: "... the MCY function is the best function among ..." a) best in which respect and b) why is it the best?

P14: There are two Zhou et al., 2015 indicate in the text to which you are referencing to

Figure 3 and in text: I recommend to use a different symbol for water vapor than P which is precipitation. It may be also useful to consider physical units of the quantities within the derivation.

---

## Referee Comment (RC2) · Anonymous Referee #2 · 3 Jan 2019

This manuscript claims to give a new, "more physical" derivation of the MCY model of the Budyko equation describing long-term catchment evapotranspiration. The authors attempt to make an analogy between water flow from one catchment to another via transport in the atmosphere, followed by precipitation in the receiving catchment, and Ohm's law as applied to two resistors arranged in series in an electrical circuit. No justification based on catchment hydrology is given for this analogy or for the three assumptions and two corollaries that it is purportedly derived from. The authors' resulting "Ohm's law expression" (Equation 18) is then converted into the MCY model by making another assumption, that the functional form of each term is a power-law in the independent variable. The manuscript also contains many erroneous statements (for example, that homogeneity is the basis for Equation 19). Experts on modeling the

Budyko equation will find nothing "new" or "more physical" in this manuscript.

---

## Short Comment (SC1) · 9 Jan 2019

The paper aims for a physically based derivation of the mean water and energy balance equation. The authors use a description of water vapor transfer between catchments and describe the fluxes in a flux gradient approach. Imposing the Budyko hypothesis then yields the well known Mezentsev-Choudhury-Yang equation. I think that a general derivation of the Budyko or the MCY equation is of high interest for hydrological research and thus of interest for HESS. However, one of aims of this paper is a rigorous derivation which reflects hydrological understanding. To be honest, I find it difficult to understand the reasoning which form the basis for the derivation. What I do not understand is the framework of water vapor transfer between catchments (illustrated in the figures). Figure 1 and 2 show a moisture transfer from one catchment to the next

(downwind?) where the input of the next catchment is set by the evaporation of the first. This then leads to their statement that after n catchments there is no water left and E = 0. However, in reality there are also other sources of vapor which can contribute to precipitation in catchment 2 and these are being neglected. So this framework is not intuitive to me. The main derivation is illustrated with figure 3 where 4 different nodes are introduced. With corollary 2 it is stated that the resistance nAB 6= n1 arguing that there are other possible routes between the node Atmosphere A and the catchment node B, namely atmosphere A to atmosphere C to catchment D to catchment B. I did not understand this water transfer between the two catchments. I am also not sure if these assumptions and the ones in stated in section 2.1 are actually relevant for the derivation described in section 3. Therefore I recommend major revisions which should particularly improve the description to enable the reader to better understand how considering hydrological processes lead towards the MCY equation.

Response:

We thank the reviewer very much for taking the time to review our manuscript and for the invaluable comments. And we will carefully revise the manuscript following the comments and suggestions to enable the reader to better understand. I am sorry that we didn't describe the catchment network clearly, especially no detailed description in the figure captions. We will give more detailed explanation in the revised version. As well known, at a long time scale, the water evaporated into atmosphere will be precipitated on land due to water cycle. In our manuscript, we defined the catchment network for water (vapor) transformation and transportation. To define the catchment network, we track the water using Lagrangian particle tracking method, i.e. the water precipitated into the first catchment is our study object and we marked it as P0; we focuses on the transportation and transformation of P0 and all the catchments that the water enters into was defined as the catchment network. For a special catchment of the catchment network, some precipitation comes from P0 and the other comes from other sources. In Figure 1, Catchments $A2,j$ (j=1, 2, 3, . . . ) represent the catchments

that the evaporated water from Catchment 1 can reach; while Catchments $A3,j$ ($j$=1, 2, 3, ...) represent the catchments that the evaporated water from Catchments $A2,j$ ($j$=1, 2, 3, ...) can reach. Regarding the precipitation in Catchment 2 as the reviewer concerns, some comes from Catchment 1 and the other comes from other sources; however, according to the Lagrangian method, we only focus on the part from Catchment 1. Also, we can establish the balance equation of only the water from Catchment 1 for Catchment 2, $P2 = E2 + R2$, where $P2$ being the precipitation from Catchment 1, $E2$ and $R2$ being the evaporation and the runoff from the evaporation from Catchment 1, respectively. Figure 2 indicates the transportation and transformation of the water $P1$ according to the particle method. Accordingly, $E2$ doesn't represent all the evaporation from Catchment 2 but only the part originating from $P1$ (precipitated and evaporated from Catchment 1). Similarly, $E3$ only represents the part of evaporation originating from $P1$. For example, we assumes that $P1 = 100$ and only focuses on the transportation and transformation for the 100 water. $P1$ transforms into $E1 = 60$ and $R1 = 40$ in Catchment 1; then $E1$ possibly transforms into $E2 = 36$ and $R2 = 24$ in Catchment 2. Regarding the water balance of Catchment 2, it is possible that the precipitation is 80 (including 60 from Catchment 1) and the evaporation is 48 (including 36 from $E1$). Regarding Figure 3, both of the potential of Points A and B (water in liquid) are zero, so we assume the two points directly connected. However, the potential difference between the two points equaling zero, which leads to no flux between the points. The objective of Section 2 is to obtain Equation (18), while that of Section 3 is to yield Equation (25). Then substitution of Equation (25) into Equation (18) leads to Equation (26).

Further remarks:

abstract: "homogeneity assumption" should be described more specific to the paper End of abstract, L15: There is no conclusion provided. Please explain what your results imply.

Response:

Thanks a lot. The homogeneity assumption indicated that the generalized flux has the same form for both water vapor transportation and chase transformation. The results may imply that the homogeneity needs further test or the Budyko hypothesis should be non-homogeneity. We will give more clear explanation in the revised version.

P5L2-5: it is unclear why this is mentioned here

Response:

In this paragraph, we wanted to give a brief review on the mathematical derivation for the Budyko hypothesis.

P6L18: Garrison 2017 not in bibliography

Response:

I am sorry for our carelessness and will add the reference in the revised version.

P8L10: $\varphi(E0) = $ ; while $\varphi = E/P$ on L27 ; unclear why the symbols is used for different meanings

Response:

It was caused by our carelessness. We will use the different symbols in the revised version.

P9L1: ". . . the MCY function is the best function among . . ." a) best in which respect and b) why is it the best?

Response:

Zhou et al. (2015) concludes "Based on the form and properties, Mezentsev-Choudhury-Yang's function is a better one among the existing Budyko functions to describe the water-energy balance in the two-dimensional state space (E0/P, E/P)" (Zhou, S., B. Yu, Y. Huang, and G. Wang (2015), The complementary relationship and generation of the Budyko functions,Geophys. Res. Lett., 42, 1781–

1790,doi:10.1002/2015GL063511). In this manuscript, we found only MCY equation satisfies Equation (18). Therefore we speculate that it is a best form. It is only a speculation. We will revise the expression in the revised version.

P14: There are two Zhou et al., 2015 indicate in the text to which you are referencing to

Response:

We will revise them in the revised version.

Figure 3 and in text: I recommend to use a different symbol for water vapor than P which is precipitation. It may be also useful to consider physical units of the quantities within the derivation.

Response:

Thanks a lot. We will use a different symbol and do more analysis on the physical units in the revised version.

---

## Short Comment (SC2) · 9 Jan 2019

We really appreciate the review and comments from the reviewer.

Regarding the "more physical" derivation, I think that the derivation for Equation (18) has more physical meaning, proposing a catchment network using Lagrangian particle tracking method, establishing the equations based on the Ohm-type law, and giving the boundary conditions of catchment hydrology. According to the Ohm-type law, we assumed water vapor being forced by some potential difference, similar to the movement of soil moisture.

Regarding "new" understanding, one is that the previous derivations for the MCY equation had an underlying assumption, i.e. the homogeneous assumption, the generalized

flux having the same form for both water vapor transportation and chase transformation. It indicates that the assumption needs further test or another form is more suitable for the real catchment.

As the review pointed out, Equation (19) and the text aren't rigorous. Instead, the homogeneity assumption is included in Equation (18). We will revise the manuscript.

---

## Author Comment (AC1) · 10 Mar 2019

The paper aims for a physically based derivation of the mean water and energy balance equation. The authors use a description of water vapor transfer between catchments and describe the fluxes in a flux gradient approach. Imposing the Budyko hypothesis then yields the well known Mezentsev-Choudhury-Yang equation. I think that a general derivation of the Budyko or the MCY equation is of high interest for hydrological research and thus of interest for HESS. However, one of aims of this paper is a rigorous derivation which reflects hydrological understanding. To be honest, I find it difficult to understand the reasoning which form the basis for the derivation. What I do not understand is the framework of water vapor transfer between catchments (illustrated in the figures). Figure 1 and 2 show a moisture transfer from one catchment to the next

(downwind?) where the input of the next catchment is set by the evaporation of the first. This then leads to their statement that after n catchments there is no water left and E = 0. However, in reality there are also other sources of vapor which can contribute to precipitation in catchment 2 and these are being neglected. So this framework is not intuitive to me. The main derivation is illustrated with figure 3 where 4 different nodes are introduced. With corollary 2 it is stated that the resistance nAB 6= n1 arguing that there are other possible routes between the node Atmosphere A and the catchment node B, namely atmosphere A to atmosphere C to catchment D to catchment B. I did not understand this water transfer between the two catchments. I am also not sure if these assumptions and the ones in stated in section 2.1 are actually relevant for the derivation described in section 3. Therefore I recommend major revisions which should particularly improve the description to enable the reader to better understand how considering hydrological processes lead towards the MCY equation.

Response:

We thank the reviewer very much for taking the time to review our manuscript and for the invaluable comments. And we will carefully revise the manuscript following the comments and suggestions to enable the reader to better understand. I am sorry that we didn't describe the catchment network clearly, especially no detailed description in the figure captions. We will give more detailed explanation in the revised version. As well known, at a long time scale, the water evaporated into atmosphere will be precipitated on land due to water cycle. In our manuscript, we defined the catchment network for water (vapor) transformation and transportation. To define the catchment network, we track the water movement using Lagrangian particle tracking method, i.e. we took the water precipitated into the first catchment as research object and marked it as P1; we focuses on the subsequent transportation and transformation of P1 and all the catchments that the water enters into was defined as the catchment network. Remarkably, for a special catchment of the catchment network, part of precipitation comes from P0 and the rest comes from other sources, and we only studied the former. In Figure 1, Catchments A2,j (j=1, 2, 3, . . . ) represent all the catchments that the evaporated water from Catchment 1 can fall down with a form of precipitation, where water vapor has been through once of evaporation-precipitation process from P1; while Catchments A3,j (j=1, 2, 3, . . . ) represent the catchments that the evaporated water from Catchments A2,j (j=1, 2, 3, . . . ) can fall down with a form of precipitation, where water vapor has been through twice of evaporation-precipitation process from P1. Regarding the precipitation in Catchment 2 as the reviewer concerns, some comes from Catchment 1 and the rest comes from other sources; however, according to the Lagrangian particle tracking method, we only focus on the part from Catchment 1. Also, we can establish the balance equation of only the water from Catchment 1 for Catchment 2, P2 = E2 + R2, where P2 being the precipitation from Catchment 1, E2 and R2 being the evaporation and the runoff from the evaporated water from Catchment 1, respectively. Figure 2 indicates the transportation and transformation of the water P1 according to the Lagrangian particle tracking method. Accordingly, E2 doesn't represent all the evaporation from Catchment 2 but only the part originating from P1 (precipitated and evaporated from Catchment 1). Similarly, E3 only represents the part of evaporation originating from P1. For example, we assumes that P1 = 100 and only focuses on the transportation and transformation for the 100 water. P1 transforms into E1 = 60 and R1 = 40 in Catchment 1; then E1 possibly transforms into E2 = 36 and R2 =24 in Catchment 2. Regarding the water balance of Catchment 2, it is possible that the precipitation is 80 (including 60 from Catchment 1) and the evaporation is 48 (including 36 from E1). Figure 2 will be deleted since Figure 1 has shown the relationship. Regarding Figure 3 (Figure 2 in the revised version), both of the potential of Points A and B (water in liquid) are zero, so we assume the two points directly connected to simplify this catchment network. It neglected the potential differences caused by other factor, such as elevation, temperature. However, the potential difference between the two points equaling zero, which leads to no flux between the points. In this manuscript, the objective of Section 2 is to obtain Equation (18), while that of Section 3 is to yield Equation (25) (equation 23 in the revised version). Then substitution of Equation (25)

into Equation (18) leads to Equation (26) (equation 24 in the revised version).

Further remarks:

abstract: "homogeneity assumption" should be described more specific to the paper End of abstract, L15: There is no conclusion provided. Please explain what your results imply.

Response:

Thanks a lot. The homogeneity assumption indicated that the generalized flux has the same form for both water vapor transportation and chase transformation, or precipitation and potential evaporation have an equalized effect on evaporation. The results may imply that the homogeneity needs further test or the Budyko hypothesis should be non-homogeneity. We will give more clear explanation in the revised version, and the revised text as given as: "The Budyko hypothesis has been widely used to describe precipitation partitioning at the catchment scale. Many empirical and analytical formulas have been proposed to describe the Budyko hypothesis. Based on dimensional analysis and mathematic reasoning, previous studies have given an analytical derivation, i.e., the Mezentsev-Choudhury-Yang (MCY) equation. However, few hydrological processes are involved in the derivation. Therefore, this study firstly defined a catchment network to describe water vapor transformation and transportation using the Lagrangian particle tracking method; and then defined the generalized flux of water vapor, which can be expressed as the ratio of potential difference with resistance. Furthermore, this study gave a new derivation of the Budyko hypothesis based on an analogy of the Ohms-type approach and the homogeneity assumption, i.e., the generalized flux has the same form for both water vapor transportation and chase transformation, and in other words, precipitation and potential evaporation have an equalized effect on evaporation. The derived equation has the same form as the MCY equation but has a more physical explanation than the mathematic reasoning proposed in previous studies. In addition, this study suggested a more general expression $E=P(b+kE_0)/(P^n+(b+kE_0)^n)^{(1/n)}$ under conditions without the homogeneity constraint, where $E$, $E_0$ and $P$ are evaporation, potential evaporation and precipitation, respectively, and $n$, $k$ and $b$ are constants. Setting b to 0, $E=(kE_0*P)/(P^n+(kE_0)^n)^{(1/n)}$ can be obtained, which was proposed by Zhou et al. (2015). It is the MCY equation when $b = 0$ and $k = 1$."

P5L2-5: it is unclear why this is mentioned here

Response:

In the original manuscript, we tried to interpret the generalized function by giving some examples. However, we found that it isn't suitable for this section and prepared to move it into discussion section.

P6L18: Garrison 2017 not in bibliography

Response:

I am sorry for our carelessness and will revise it in the revised version.

P8L10: $\varphi(E_0)$ = ; while $\varphi = E/P$ on L27 ; unclear why the symbols is used for different meanings

Response:

It was caused by our carelessness. We will use the different symbols in the revised version.

P9L1: ". . . the MCY function is the best function among . . ." a) best in which respect and b) why is it the best?

Response:

Zhou et al. (2015) concludes "Based on the form and properties, Mezentsev-Choudhury-Yang's function is a better one among the existing Budyko functions to describe the water-energy balance in the two-dimensional state space (E0/P, E/P)"

(Zhou, S., B. Yu, Y. Huang, and G. Wang (2015), The complementary relation-ship and generation of the Budyko functions,Geophys. Res. Lett., 42, 1781–1790,doi:10.1002/2015GL063511). In this manuscript, we found only MCY equation satisfies Equation (18). Therefore we speculate that it is a best form. It is only a speculation. We will remove it in the revised version.

P14: There are two Zhou et al., 2015 indicate in the text to which you are referencing to

Response:

We will revise them in the revised version.

Figure 3 and in text: I recommend to use a different symbol for water vapor than P which is precipitation. It may be also useful to consider physical units of the quantities within the derivation.

Response:

Thanks a lot. We will use a different symbol (V) and do more analysis on the physical units in the revised version. The unites of fluxes (P, E, E0) are mm/a. We will emphasize the units within the derivation in the revised version.

---

## Author Comment (AC2) · 10 Mar 2019

This manuscript claims to give a new, "more physical" derivation of the MCY model of the Budyko equation describing long-term catchment evapotranspiration. The authors attempt to make an analogy between water flow from one catchment to another via transport in the atmosphere, followed by precipitation in the receiving catchment, and Ohm's law as applied to two resistors arranged in series in an electrical circuit. No justification based on catchment hydrology is given for this analogy or for the three assumptions and two corollaries that it is purportedly derived from. The authors' resulting "Ohm's law expression" (Equation 18) is then converted into the MCY model by making another assumption, that the functional form of each term is a power-law in the independent variable. The manuscript also contains many erroneous statements

Creative Commons CC-BY license logo

(for example, that homogeneity is the basis for Equation 19). Experts on modeling the Budyko equation will find nothing "new" or "more physical" in this manuscript.

Response:

We really appreciate the reviewer taking time to review our manuscript and giving important comments.

Regarding the "more physical" derivation, I think that the derivation for Equation (18) has more physical meaning, namely proposing a catchment network using Lagrangian particle tracking method, establishing the equations based on the Ohm-type law, and giving the boundary conditions of catchment hydrology. According to the Ohm-type law, we assumed water vapor being forced by some potential difference, similar to the movement of soil moisture.

Regarding "new" understanding, one is that the previous derivations for the MCY equation had an underlying assumption, i.e. the homogeneous assumption, the generalized flux having the same form for both water vapor transportation and chase transformation, and in other words precipitation and potential evaporation having an equalized effect on evaporation. It indicates that the assumption needs further test or another form is more suitable for the real catchment.

As the review pointed out, Equation (19) and the text aren't rigorous. Instead, the homogeneity assumption is included in Equation (18). We will revise the manuscript.

In addition, we have revised the conclusion as: "Previous studies have analytically derived the mean annual water-energy balance equation for the Budyko hypothesis mainly by mathematical reasoning, such as Fu (1981), Yang et al. (2008), and Zhou et al. (2015). To give a new derivation with more physical meaning, this study focused on subsequent transportation and transformation of the precipitation fallen down to a certain catchment using the Lagrangian particle tracking method, proposed a catchment network in which water vapor was transformed and transported through evaporation-

precipitation processes, defined the generalized flux of water vapor, and expressed the generalized flux as the ratio of potential difference with resistance by using an Ohms-type approach. On the base of these, the relationship among potential evaporation ($E0$), precipitation ($P$) and evaporation ($E$), $1/(f(E))=1/(f(E_0))+1/(f(P))$, was achieved, in which $f()$ represents the generalized flux (i.e. a function of flux). Furthermore, the MCY equation $E=(PE_0)/((P^n+E_0^n))^{(1/n)}$ was derived based on mathematic reasoning. Remarkably, an implicit homogeneity assumption for the MCY equation was exposed, i.e., the generalize function has the same form for both vapor transportation and chase transition, and in other words, precipitation and potential evaporation have an equalized effect on evaporation. In addition, without the homogeneity assumption, this study suggested a general form $E=P(b+kE_0)/(P^n+(b+kE_0)^n)^{(1/n)}$, where b and k are constants. The equation can be simplified to $E=((kE_0*P)(P^n+(kE_0)^n)^{(1/n)}$ proposed by by Zhou et al. (2015) if setting $b = 0$; and the MCY equation if setting $b = 0$ and $k = 1$."